# Exploring the Role of Explainability for Uncovering Bias in Deep Learning-based Medical Image Analysis

**Emma A.M. Stanley**[1,2,3,4]                                         EMMA.STANLEY@UCALGARY.CA
**Matthias Wilms**[2,4,5,6]                                          MATTHIAS.WILMS@UCALGARY.CA
**Pauline Mouches**[1,2,3,4]                                        PAULINE.MOUCHES@UCALGARY.CA
**Nils D. Forkert**[1,2,3,4,7]                                         NILS.FORKERT@UCALGARY.CA

[1] *Department of Radiology, University of Calgary, Canada*

[2] *Hotchkiss Brain Institute, University of Calgary, Canada*

[3] *Department of Biomedical Engineering, University of Calgary, Canada*

[4] *Alberta Children's Hospital Research Institute, University of Calgary, Canada*

[5] *Department of Pediatrics, University of Calgary, Canada*

[6] *Department of Community Health Sciences, University of Calgary, Calgary, Canada*

[7] *Department of Clinical Neurosciences, University of Calgary, Calgary, Canada*

## Abstract

Fairness and bias are critical considerations for the effective and ethical use of deep learning models for medical image analysis. Despite this, there has been minimal research on how explainable artificial intelligence (XAI) methods can be leveraged to better understand underlying causes of bias in medical image data. To study this, we trained a convolutional neural network on brain magnetic resonance imaging (MRI) data of 4547 adolescents to predict biological sex. Performance disparities between White and Black racial subgroups were analyzed, and average saliency maps were generated for each subgroup based on sex and race. The model showed significantly higher performance in correctly classifying White males compared to Black males, and slightly higher performance for Black females compared to White females. Saliency maps indicated subgroup-specific differences in brain regions associated with pubertal development, an established confounder in this task, which is also associated with race. These findings suggest that models demonstrating performance disparities can also lead to varying XAI outcomes across subgroups, offering insights into potential sources of bias in medical image data.

**Keywords:** Fairness, bias, explainability

## 1. Introduction

Recently, there has been growing concern about bias and fairness issues in deep learning models for medical image analysis. Prominent examples have included observed performance disparities between different sociodemographic groups in chest X-ray classification (Seyyed-Kalantari et al., 2021) and cardiac segmentation (Puyol-Antón et al., 2021). Although many studies have shown that deep learning models can produce disparate outcomes, little research has been done to understand the root cause of biases related to performance and how they manifest in such models. While explainable AI (XAI) methods have been commonly applied to understand black-box deep learning model decisions, they have not been used extensively to better understand bias and fairness in medical imaging models. This study aims to evaluate if XAI is a feasible technique for better understanding potential

sources of bias that result in subgroup-specific performance disparities, using a well-defined deep learning classification task. More precisely, a convolutional neural network (CNN) is trained to classify biological sex of adolescents, in which a previous study identified the stage of pubertal development as a significant confounding factor (Adeli et al., 2020). Due to established differences in the onset of pubertal development between different races and sexes (Herman-Giddens et al., 2012; Wu et al., 2002), we hypothesized that this model could produce performance disparities between Black and White subgroups. We posit that XAI could then provide clues to sources of bias in medical imaging data if brain regions associated with the known confounder of pubertal development are identified as salient for the model's predictions. This short paper summarizes the study in (Stanley et al., 2022b).

## 2. Methods

This study used T1-weighted brain MRI from 4547 participants aged 9-10 from the 3.0 release of the ABCD study[1]. The biological sex (defined as sex assigned at birth) and race information of the participants were collected from surveys completed by a parent or guardian. Out of the total number of participants, 3,008 were identified as White and 390 were identified as Black. The remaining participants of other races were included in model training, but not in subgroup analyses. A CNN based on the Simple Fully Convolutional Network proposed by (Peng et al., 2021) was used for the sex classification task, with a five-fold cross-validation scheme. Full model implementation details are available in (Stanley et al., 2022b). Saliency maps were computed by averaging registered SmoothGrad (Smilkov et al., 2017) results from 20 correctly classified subjects within each demographic subgroup. Weighted saliency scores were computed by multiplying the percent of salient voxels within each brain region defined by the CerebrA atlas (Manera et al., 2020) by a weighting factor accounting for the mean saliency intensity value within each region. To evaluate differences in model performance between White and Black subgroups, a two-tailed Student's t-test with a significance level of 0.05 was used.

## 3. Results and Discussion

The sex classification model achieved an overall accuracy of 87.8%, comparable to the results reported by (Adeli et al., 2020). While classification accuracy within the White female subgroup was lower but not statistically significantly different from the Black female subgroup (86.5% vs. 89.3%, p=0.260), the White male subgroup accuracy was 9.2% higher than Black male subgroup accuracy, which was significant (90.3% vs. 81.1%, p=0.03). These results, similar to those reported by (Seyyed-Kalantari et al., 2021) and (Puyol-Antón et al., 2021), highlight the importance of not relying solely on high overall accuracy to evaluate model performance, as disparities may exist within sensitive subgroups and should be reported. Although this study used race as a grouping factor, a major challenge for evaluating model fairness is that other hidden disparities may be present within sensitive attributes not explicitly analyzed, or within intersections of sensitive groups (Stanley et al., 2022a).

---

1. https://abcdstudy.org/

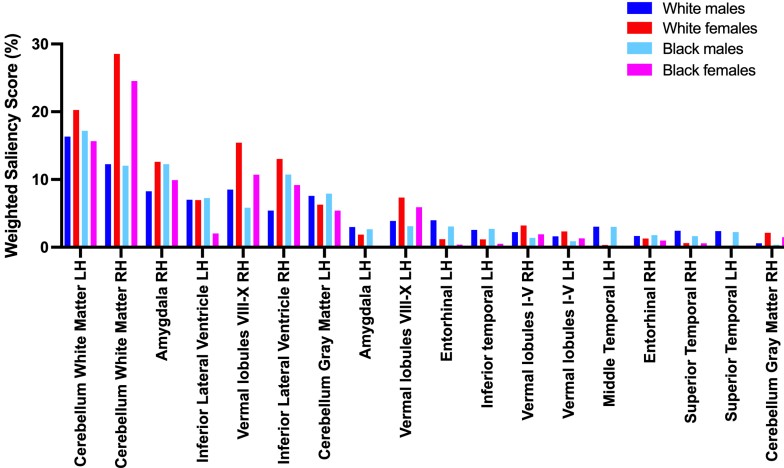

Figure 1: Weighted saliency scores in top brain regions (RH = right hemisphere, LH = left hemisphere).

Brain regions highlighted in the saliency maps included the cerebellum, amygdala, lateral ventricles, temporal lobes, and entorhinal cortex, with the cerebellum showing the highest saliency activation. This region was also identified as the most significant confounder related to pubertal development stage for sex classification in (Adeli et al., 2020). The weighted saliency scores for each subgroup are presented in Fig 1, with some brain regions showing differences between sexes and races. For example, the right hemisphere (RH) cerebellum white matter and left hemisphere (LH) cerebellum gray matter show sex-specific trends, and the RH vermal lobules VIII to X and LH entorhinal cortex show sex-specific trends by race. The amygdala and medial temporal lobe, which have been linked to morphological changes associated with pubertal development (Bramen et al., 2011), also demonstrate subgroup differences in saliency scores. These varying saliency scores within brain regions may be due to the model using morphological information related to pubertal development stage differently for each subgroup, potentially contributing to performance disparities. While subgroup saliency maps may help link model performance to bias and confounders in datasets, it should also be noted that these results have implications on the use of XAI for biomarker detection in clinical tasks. If saliency maps show appreciable differences between demographic groups, conclusions based on aggregate saliency maps may not be generalizable to distinct subpopulations.

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
