# OpenReview forum: "Exploring the Role of Explainability for Uncovering Bias in Deep Learning-based Medical Image Analysis"
_MIDL.io/2023/Short_Paper_Track — MIDL 2023 Short paper track Poster_

### Official Review · Reviewer_1HLX · 2023-04-12
**Interesting discussions in perspective**

**Rating:** 8
**Confidence:** 4

**Review:**

This submission summarises the work of the authors published in JMI that explores whether explainability approaches can help identify biases in classification models. The summary is clear. The topic is important and should lead to interesting discussions.

---

### Official Review · Reviewer_gDxV · 2023-04-21
**Short paper review**

**Rating:** 7
**Confidence:** 3

**Review:**

The paper investigates saliency maps to better understand sources of bias in medical image classification. The focus is classification of biological sex in brain MR images, training on 4537 participants, of which 3,008 were identified as White and 390 were identified as Black. The results show a significant different in accuracy between White and Black males, and the saliency maps are reported to identify brain regions that are associated both with sex and race.

Pros:
* Important topic to investigate
* Results appear to reflect prior knowledge about brain development

Cons:
* Imbalanced data, unclear how this affects model performance
* Unclear why only 20 correctly classified subjects are used for saliency map creation

Suggestions:
* The work of Gichoya et al - AI recognition of patient race in medical imaging: a modelling study could be relevant to include.
* Other evaluation metrics besides accuracy could be relevant